# Intra- and Interrater Reliability of CT- versus MRI-Based Cochlear Duct Length Measurement in Pediatric Cochlear Implant Candidates and Its Impact on Personalized Electrode Array Selection

**DOI:** 10.3390/jpm13040633

**Published:** 2023-04-04

**Authors:** Jan Peter Thomas, Hannah Klein, Imme Haubitz, Stefan Dazert, Christiane Völter

**Affiliations:** 1Department of Otorhinolaryngology, Head and Neck Surgery, St. Johannes Hospital, Cath. St. Paulus Society, Academic Teaching Hospital of the University of Münster, Johannesstr. 9-17, 44137 Dortmund, Germany; 2Department of Otorhinolaryngology, Head and Neck Surgery, Katholisches Klinikum, Ruhr University Bochum, Bleichstr. 15, 44787 Bochum, Germany

**Keywords:** cochlear implant, lateral wall electrode array, cochlear duct length, radiological measurement, CT, MRI, personalized electrode array, angular insertion depth

## Abstract

Background: Radiological high-resolution computed tomography-based evaluation of cochlear implant candidates’ cochlear duct length (CDL) has become the method of choice for electrode array selection. The aim of the present study was to evaluate if MRI-based data match CT-based data and if this impacts on electrode array choice. Methods: Participants were 39 children. CDL, length at two turns, diameters, and height of the cochlea were determined via CT and MRI by three raters using tablet-based otosurgical planning software. Personalized electrode array length, angular insertion depth (AID), intra- and interrater differences, and reliability were calculated. Results: Mean intrarater difference of CT- versus MRI-based CDL was 0.528 ± 0.483 mm without significant differences. Individual length at two turns differed between 28.0 mm and 36.6 mm. Intrarater reliability between CT versus MRI measurements was high (intra-class correlation coefficient (ICC): 0.929–0.938). Selection of the optimal electrode array based on CT and MRI matched in 90.1% of cases. Mean AID was 629.5° based on the CT and 634.6° based on the MRI; this is not a significant difference. ICC of the mean interrater reliability was 0.887 for the CT-based evaluation and 0.82 for the MRI-based evaluation. Conclusion: MRI-based CDL measurement shows a low intrarater difference and a high interrater reliability and is therefore suitable for personalized electrode array selection.

## 1. Introduction

Cochlear implantation is the treatment of choice in people with severe to profound hearing loss who derive an insufficient benefit from hearing aids [1,2]. Although most cochlear implant (CI) users obtain a good speech outcome, the extent of speech perception varies [3]. Duration of deafness and of hearing aid use, age at implantation, extent of residual hearing, and other personal factors, such as neurocognitive abilities, have been recognized as underlying factors in postoperative outcomes [4,5,6,7,8,9].

The type of electrode array and its intracochlear position might also impact postoperative outcomes. Lateral wall electrode arrays which stimulate the peripheral processes of the spiral ganglion cells have to be differentiated from precurved perimodiolar and mid-scala arrays, both of which directly stimulate the spiral ganglion cells [10]. However, it is still unclear to what extent differences in the aforementioned designs affect postoperative speech understanding [11]. No significant difference was observed in postoperative speech perception [7,11] or in the consonant-nucleus-consonant (CNC) score 24 months postoperatively [12] between lateral wall and perimodiolar electrode arrays of three major CI manufacturers (Advanced Bionics^®^, Stäfa, Switzerland, Cochlear^®^, Sydney, Australia, and MED-EL^®^, Innsbruck, Austria). However, studies have shown that in precurved electrode arrays, modiolar proximity positively affects postoperative speech outcomes, whereas in lateral wall electrode arrays, angular insertion depth (AID) is a highly significant factor for postoperative speech perception [13].

As spiral ganglion cell bodies have been found beyond the basal turn of the human cochlea, coverage of more apical parts of the cochlea by lateral wall electrodes might be useful to stimulate as many ganglion cells as possible, including those encoding lower frequencies [14]. Buchman et al. compared a medium-length electrode array (24 mm) and a standard-length lateral wall array (31.5 mm) from MED-EL^®^, Innsbruck, Austria, and showed a trend towards better speech perception with longer electrode arrays and a greater AID in a prospective study in 13 CI users. Statistical significance was shown in a retrospective evaluation in a larger group of 19 CI users [15]. This result complies with Canfarotta et al., who reported a significantly better speech perception in the long-term follow-up 4 years after implantation in subjects implanted with an array of 31.5 mm length compared to those implanted with a shorter medium array [16]. Further studies comparing lateral wall electrode arrays of different lengths (MED-EL^®^, Innsbruck, Austria FLEX 24 with a median AID of 408°, FLEX 28 with a median AID of 575°, and STANDARD with a median AID of 584°) in 48 subjects clearly demonstrated that a greater AID directly correlates with improved postoperative speech perception, as assessed by CNC score [17]. Further, by comparing implants of all three manufacturers, a greater AID of the electrode array positively correlated with improved postoperative monosyllabic word recognition scores; the CNC score increased by 0.6% with every 10-degree increase in the AID [7]. Morrel et al. confirmed this observation in a study that compared lateral wall electrode arrays of the 3 manufacturers in 177 ears [18].

In addition to beneficial postoperative speech perception in quiet, a deeper insertion of lateral wall electrode arrays has shown better pitch estimation [19]; a reduction in frequency-to-place mismatch [16]; and an improvement in low-frequency perception, which results in higher satisfaction in CI users when listening to music [20].

However, the size of human cochleae is highly variable, with a range from 25 to 45 mm in the cochlear duct length (CDL), as shown in numerous histological and radiological measurements [21,22,23,24]. Recently, Cooperman et al. demonstrated a significant association between the electrode-to-CDL ratio (ECDLR) and the audiological outcomes in CI users implanted with lateral wall electrodes [25]. To achieve complete cochlear coverage, lateral wall electrode arrays are available in various lengths (20, 24, 26, 28, and 31.5 mm arrays), thereby allowing clinicians to select an array best suited to each candidate’s CDL [26,27]. To this end, highly accurate preoperative measurement of the CDL is desirable.

Currently, preoperative CDL measurement is usually performed radiologically by using high-resolution computed tomography (HR-CT) data of the temporal bone. For this, different options are available [28]: direct measurements of the CDL by projecting the cochlear duct onto a 2D plane [29] and using contouring techniques [30]; 3D reconstructions of the cochlea [26]; measurement of spiral coefficients with cochlear parameters calculating the CDL [31,32]. However, correct CDL measurement is not feasible in candidates with any type of cochlear dysplasia.

For clinical use, the tablet-based Otoplan software (CAScination AG, Bern, Switzerland in collaboration with MED-EL^®^, Innsbruck, Austria) was developed to facilitate and speed up CDL measurement as a standard preoperative procedure. Based on the principles established by Alexiades et al., who calculated the length of the cochlea by the length of the diameter of the basal cochlea [31], Otoplan (version 2.0) calculates the CDL according to an elliptic-circular approximation (ECA) method [33]. ECA is based on the two basal turn parameters (diameter and width) which yield more accurate predictions of the CDL than the frequently adopted Escudé approach [34] which relies on the measurement of one single diameter [35].

The ECA method used in non-malformed cochleae provides data that are similar or even more accurate to those obtained by direct measurement using a curved multiplanar reconstruction method for 3D reconstruction. Chen et al. used Otoplan version 2.0 and found no significant difference in the CDL, the diameters A and B, or the height H between both methods, whereas time of measurement significantly decreased to 5.9 ± 0.69 min by using the Otoplan system compared to 9.3 ± 0.72 min by using the curved multiplanar reconstruction method. In addition, the intraindividual reliability of repetitive measurements using the same evaluation method was better with Otoplan than with the curved multiplanar reconstruction method [32]. Breitsprecher et al. compared Otoplan version 3.0 to a specifically designed preclinical 3D reconstruction software and to the established A-value method. They found that Otoplan provided more accurate CDL measurements; hereby measurement of the width of the cochlea (B-value) had a larger influence on the determination of the CDL than the diameter (A-value) [36,37].

Currently, both HR-CT and MRI of the skull/temporal bone are the standard diagnostic procedures to assess the individual anatomy of the cochlea and the auditory nerve prior to cochlear implantation [38].

However, the impact of ionizing X-rays, especially on young children, has to be taken into account. In 2012, Pearce et al. published a well-respected study showing that CT scans with a cumulative dose of about 50 mGy might almost triple the risk of leukaemia and a dose of about 60 mGy might triple the risk of brain tumours in children aged less than one year [39]. The radiation dose of one single cranial CT is capable of inducing a thyroid or brain tumour in infants or children and therefore has a high impact on life expectancy and quality of life [40]. This has been confirmed by recent studies in young subjects after low-dose radiation exposure with a statistically significant dose–response relationship for central nervous system tumours and leukaemia after CT scans at this age [41,42]. Therefore, when considering HR-CT of the temporal bone in very young and young children prior to cochlear implantation, clinicians must carefully weigh its advantages and disadvantages.

Today, some institutions routinely use MRI as the sole modality of preoperative imaging in the assessment of children referred for cochlear implantation [43]. Recently, in their study on children aged less than 36 months, Ehrmann-Mueller showed that cochlear implantation without preoperative CT but only MRI is a safe procedure without additional risks in young and very young children [44].

For these patients obtaining reliable CDL measurement based on MRI data is necessary for choosing the optimal electrode array length in people scheduled for cochlear implantation with a lateral wall electrode.

Therefore, the present study aimed to (1) compare CDL measurement with CT and CDL measurement with MRI, (2) analyse the impact of the different radiological methods on the selection of personalized electrode array length, and (3) study interrater differences of three different raters on CDL measurements in children with congenital bilateral profound hearing loss who were implanted at an early age.

## 2. Materials and Methods

### 2.1. Participants

All children with bilateral severe to profound hearing loss scheduled for cochlear implantation between 11/2015 and 09/2020 were screened for inclusion in the study. Inclusion criteria were (1) age of <18 years, (2) availability of high-quality HR-CT data and high-resolution temporal bone/cranial MRI with a slice thickness of 0.6 mm, including T2-weighted 3D-CISS sequences, and (3) absence of anatomical malformations of the inner ear or of the temporal bone. All participants underwent radiological examinations (HR-CT and MRI) at the Departments of Radiology at the Catholic Hospital Bochum in Germany.

### 2.2. Radiological Data

HR-CT data of the temporal bone were obtained using the Siemens Somatom Emotion 16 tube voltage 110 kV, CareDose, RotationTime 1.0, Collimation 4 × 0.6 mm, Pitch 0.75 or the Siemens Definition AS+ tube voltage 180 kV, Rotation Time 1.0, Collimation 128 × 0.6 mm, Pitch 0.8 (Siemens Healthineers, Erlangen, Germany).

MRI data were obtained using the Siemens Avanto DOT 1.5 T, T2w CISS 3D, time of repetition = 5.44 ms, time of echo = 2.44 ms, and field-of-view = 180 mm (Siemens Healthineers, Erlangen, Germany).

### 2.3. Measurement of Cochlear Parameters

Biometric measurement of the cochlea was performed using a tablet-based planning software (Otoplan, version 2.0, CAScination AG, Bern, Switzerland in collaboration with MED-EL^®^, Innsbruck, Austria). This included (1) the measures of the diameter of the basal turn from the round window crossing the modiolus to the opposite wall (A-value), (2) the orthogonal width of the basal turn (B-value), (3) the height of the cochlea (H-value), and (4) the total CDL calculated based on the diameter A and the width B (Figure 1). The parameters of both cochleae were evaluated in each participant.

The most appropriate electrode array length for each individual was chosen based on a recommendation by Mistrik and Jolly, who posited that the optimal length of lateral wall electrode arrays provides a coverage of 1.5–2 turns or 80% of the cochlea [45]. Therefore, we evaluated the CDL measured at 720° (two turns) for the selection of the most appropriate lateral wall electrode array for each participant. The most appropriate electrode array of the currently available lateral wall electrode portfolio by MED-EL^®^, Innsbruck, Austria (FLEX SOFT/Standard (31.5 mm), FLEX 28 (28 mm), FLEX 26 (26 mm), FLEX 24 (24 mm), or FLEX 20 (20 mm)) was then individually chosen and the AID of the most apical electrode contact no. 1 was evaluated to analyse the degree of stimulability of the low pitches.

The radiological measures of the cochlear parameters were performed by three different raters (CV, HK, JPT), independently from each other. In the first step, these measures were based on the CT data of all participants; in the second step, these measures were based on the MRI data of all participants with the rater not knowing the CT data.

To obtain sufficient experience in performing CDL measurement based on MRI data compared to CT data, an intensive training was attended by all three raters before the study. During the training, the raters obtained and evaluated cochlear parameters using comparative CT-based and MRI-based measurement in at least five subjects not included in the present study.

### 2.4. Data Analysis

Descriptive statistics including mean and standard deviation (SD) as well as ranges were used to analyse the different cochlear parameters (diameter A and B and height H). CDL and the length of the cochlea at 720° were calculated using the ECA method as described by Schurzig [33]. To determine the mean differences, a serial t-test was applied. To compare the intra- and inter-rater reliability measures based on the CT and MRI data with continuous data, intra-class correlation coefficient (ICC) for absolute agreement with a 95% confidence interval was used. Here, ICC values were interpreted as follows: poor (<0.40), fair (0.40–0.59), good (0.60–0.74), and excellent (0.75–1.00) [46]. To assess differences in AID of the most appropriate individually fitted electrode array, serial comparison of the data using the Wilcoxon–Mann–Whitney U test was performed. Differences with a *p* value < 0.05 were considered statistically significant. The statistical program used was Medas (Grund company, Margretshöchheim, Germany).

## 3. Results

An amount of 78 cochleae in 39 children (23 male, 16 female) with a mean age of 6.8 ± 5.0 years were included in the study. A total of 7 children were excluded due to anatomical malformations of the inner ear.

### 3.1. Absolute CT- and MRI-Based Length Parameters (Mean and Range)

Mean values and SDs of CT-based and MRI-based measures of the CDL, length of the cochlea at 720°, diameter A, width B, and height H separated for each side and each rater are shown in Table 1.

Comparing CT-based and MRI-based measures showed significant differences in the height H of both ears (*p* = 0.0002 or 0.0000). All other length parameters did not significantly differ between CT-based and MRI-based data (*p* ≥ 0.077).

### 3.2. Distribution of the Individual CDLs and Lengths at 720° Based on CT Data versus MRI Data

Distribution of the individual CDLs and lengths at 720° of all cochleae based on CT or MRI data are shown in Figure 2. Participants are listed separately for both sides in ascending order of the CDL mean values obtained from the CT-based data.

The CDL of the right cochleae ranged between 30.5 and 39.3 mm (mean: 34.439 ± 1.670 mm) in the CT-based measurements, and between 28.7 mm and 39.2 (mean: 34.541 ± 1.621 mm) in the MRI-based measurements. The CDL of the left cochleae ranged between 30.2 and 39.8 mm (mean: 34.375 ± 1.796 mm) in the CT-based measurements and between 31.0 mm and 39.4 mm (34.386 ± 1.662mm) in the MRI-based measurements.

The length at 720° of the right cochleae was between 28.3 mm and 36.4 mm in the CT-based measurements and between 28.7 and 36.4 mm in the MRI-based measurements. The length at 720° of the left cochleae ranged between 28.0 and 37.0 mm in the CT-based measurements and between 28.8 and 36.6 mm in the MRI-based measurements.

### 3.3. Absolute Intrarater Differences and Intrarater Reliability According to the Comparison of CT-Based Data and MRI-Based Data

Mean absolute intrarater difference and SD of the CDL/length at 720° between CT-based and MRI-based data were 0.579 ± 0.532 mm/ 0.532 ± 0.496 mm for the right cochlea and 0.477 ± 0.434 mm/0.444 ± 0.422 mm for the left cochlea. Absolute intrarater difference of the CDL/length at 720° ranged between 0 and 3.1 mm/0 and 3.0 mm for the right and between 0 and 3.2 mm/0 and 3.1 mm for the left cochlea. Absolute differences of all measured and calculated cochlear length dimensions are shown in Figure 3.

Intrarater reliability for CT-based versus MRI-based CDL measures evaluated by intra-class correlation (ICC) was 0.929 in the right cochlea and 0.938 in the left cochleae. Intrarater reliability for the length at 720° was 0.929 in the right cochleae and 0.936 in the left cochleae. Intrarater reliability for diameter A was 0.859 in the right ears and 0.890 in the left ears. Intrarater reliability for diameter B was 0.893 in the right ears and 0.914 in the left ears. Intrarater reliability for height H was 0.771 in the right cochleae and 0.754 in the left cochleae.

CT-based and MRI-based measures of height H significantly differed on both sides (*p* ≤ 0.0002). However, intrarater reliability by ICC was high (0.771 on the right and 0.754 on the left cochlea).

### 3.4. Most Appropriate Electrode Arrays (FLEX 28 vs. 31.5 mm Electrode Array) Selected by the Three Raters Based on the CT Data versus MRI Data

The most appropriate lateral wall electrode array for complete cochlear coverage was selected based on the length calculated at 720°. In the present study, all cochleae analysed had a length of at least 28.0 mm at 720° in both the CT scans and in MRI scans. Therefore, a FLEX 28 or a 31.5 mm electrode array may be the array of choice for complete cochlear coverage.

The electrode arrays (FLEX 28 vs. 31.5 mm electrode array) selected by each rater based on CT data vs. MRI data are shown in Figure 4.

Electrode arrays selected based on CT data matched those selected based on MRI data in 90.6% (106/117) of the right cochleae and in 89.74.% (105/117) of the left cochleae. There was no systematic bias in terms of a particular electrode array selection by evaluation of the cochlear parameters by CT-based vs. MRI-based data. In 67.1% of the CT-based evaluations and in 65.4% of the MRI-based evaluations of all cochleae, a 31.5 mm electrode array was selected as the most appropriate array. In the remaining 32.9% and 34.6%, a FLEX 28 electrode array was recommended.

### 3.5. AID with Personalized Electrode Arrays Evaluated on the Basis of CT or MRI Data

Mean AID of the most appropriate electrode array length (evaluated based on the individual lengths at two turns) in the left cochleae was 634.658° (range: 543.0°–699.0°) based on CT data and 631.970° (range: 535.6°–684.8°) based on MRI data. Mean AID in the right cochleae was 633.685° (range: 551.6°–701.9°) based on CT data and 629.463° (range: 555.6°–682.4°) based on MRI data. Serial comparison of the data revealed no significant difference in AID between CT-based and MRI-based data or between the three different raters (*p* = 0.42 for the left cochleae; p=0.16 for the right cochleae).

### 3.6. Absolute Interrater Differences and Interrater Reliability Based on CT Data versus MRI Data

The absolute interrater difference of the CDL/length at 720° measured by CT was 0.5538 ± 0.4354 mm/0.5162 ± 0.3967 mm for the right and 0.6547 ± 0.5140/0.605 ± 0.4648 mm for the left side. The interrater differences ranged between 0 and 2.2 mm/0 and 2.1 mm on the right and between 0 and 3.5 mm/0 and 3.1 mm on the left side.

The interrater reliability of the CDL/length at 720° measured by CT was 0.879/0.915 for the right side and 0.895/0.901 for the left side.

Absolute interrater difference of the CDL/length at 720° measured by MRI was 0.8632 ± 0.6435 mm/0.7949 ± 0.5920 mm for the right and 0.6410 ± 0.5300/0.6017 ± 0.4920 mm for the left side. The interrater difference ranged between 0 and 2.9 mm/0 and 2.7 mm on the right and between 0 and 2.8 mm/0 and 2.7 mm on the left side.

The interrater reliability of the CDL/length at 720° measured by MRT was 0.789/0.785 for the right side and 0.851/0.856 for the left side.

The number of selected electrode arrays on the left and on the right side as suggested by the three raters and the radiological method used are shown in Table 2.

Interrater consistency in electrode array selection of all three raters was achieved in 61.5% (24/39) on the left side and 66.7% (26/39) on the right side based on CT evaluation, and in 59.0% (23/39) on the left side and 61.5% (24/39) on the right side based on MRI-evaluation.

## 4. Discussion

In the present study, high interindividual variability in the size of infant cochleae was found with a CT-based CDL between 30.2 and 39.8 mm. This is in accordance with Meng et al., who found a CDL range between 30.7 and 42.2 mm in 310 subjects with non-malformed cochleae, aged 1–73 years [26], and with Timm et al., who analysed 272 cochleae with a CDL range between 31.3 and 44.9 mm in a study population without age specification [47]. It is known from previous studies that there are no age-dependent differences in CDL [24,26], as the cochlea reaches its adult size as early as 16 to 20 weeks of gestation [48]. Thus, the CDL measurements of the infant cochleae made in the present study can be compared to CDL measurements in older subjects.

In our study, calculation of the individual length at 720° by Otoplan based on the formula 0.928 × CDL showed a range between 28.0 and 37.0 mm using the CT-based data. This is similar to the outcomes by Meng et al. (range: 27.6 to 37.7 mm), who used three-dimensional multiplanar reconstructed CT for direct measurement of the length of the first two turns [26].

In the present study, mean CDL calculated from CT was 34.439 ± 1.670 mm in the right cochlea and 34.375 ± 1.796 mm in the left cochlea, which is consistent with a large number of previous studies. Müller-Graff et al. found a mean CDL of the organ of Corti (CDL_OC_) of 34.55 mm ± 1.6 mm using the same Otoplan version 2.0 with multi-slice CT data of 20 subjects aged 64 ± 14.9 years [49]. Chen et al. found a mean CDL of 34.37 ± 1.481 in 68 cochleae of 34 subjects with a mean age of 16.6 years (range: 0.6–63.6) without any statistical differences in age or side of the ear [32].

In contrast, Weber et al. reported a longer mean CDL_OC_ of 36.69 ± 1.78 mm (range CDL_OC_ 33.05–42.61 mm). These higher values might be explained by the type of Otoplan version (version 3.0) used. In version 3.0, the hook region as the part of the basilar membrane preceding the centre of the round window is factored in with 2.5 mm instead of 1.58 mm as in Otoplan^®^ version 2.0 [50]. Spiegel et al. used Otoplan version 2.0 and also reported a longer mean CDL (36.81 ± 1.8 mm) than was found in the present study [51]. However, this might probably be due to the use of the lateral wall CDL (CDL_LW_ value), which differs from the CDL_OC_ value by a factor of 0.9 [36], resulting in a mean CDL_OC_ value of 33.129 mm [51].

The mean CDL based on MRI data did not differ from those based on CT data in the present study. The mean intrarater difference between the different radiological methods was 0.579 ± 0.532 mm for the right cochleae and 0.477 ± 0.434 mm for the left cochleae. So far, only a few studies have dealt with the measurement of cochlear parameters based on MRI data. The statistically insignificant mean intrarater difference in the present study is similar to data described by Taeger et al., who analysed 42 cochleae (0.65 mm) [52]. Nash et al. also found no significant intrarater difference [43]. It is not possible to compare our data with the data obtained by George-Jones et al., who were the first to report on length measurements of the cochlea based on MRI data by Otoplan, because they did not compare the two different radiological procedures [53].

Concerning the length at 720°, there was no significant difference in diameter A and width B between the different underlying radiological procedures in the present study. In contrast, the height of the cochlea H was significantly longer in the MRI scan than in the CT scan. This has already been demonstrated by Weber et al., who also found a greater width B and a small but significantly greater mean CDL by 0.89 mm in MRI-based measurements [50]. However, our study does not confirm the suspected causes for this: the missing bone signal and a lower resolution in MRI, described in the aforementioned study. In our study, no consistently greater length parameters were detected in the MRI-based measurements than in the CT-based measurements, with the exception of the height H. The uniform setting of markers further outside, as described by Weber et al. [50], was probably avoided in the present study by an intensive comparative training of the raters prior to the study. Since the height H is not included in the formula for calculating the CDL, greater values for the height H had no influence on the CDL, or on the length at 720°, or on the selection of the optimal electrode array in the present study. However, a clear statement about the impact of differences of the resolution accuracy of the radiological procedures on the determination of the length parameters is not possible. CT has the advantage of high spatial resolution and multiplanar capabilities, whereas MRI has greater contrast resolution than CT.

Calculated intrarater reliability of the CDL and the length at two turns indicated an excellent correlation with an ICC of 0.929–0.938 in the present study. Comparative reliability data of CT-based versus MRI-based measurements of the CDL evaluated by spiral coefficients were missing so far. The only study dealing with this topic was based on measurements in three-dimensional curved multiplanar reconstructions, which also showed an excellent ICC of 0.79–0.94 [54].

Even if the mean intrarater difference between CT-based and MRI-based measurements did not show any significant difference and the intrarater reliabilities turned out to be excellent, differences in the measurement of the individual can affect the personalized selection of the electrode array. In the present study, individual intrarater differences of the CDL were at a maximum of 3 mm. This corresponds to data by Nash et al., who described a CDL difference of 0–2.92 mm between CT-based and MRI-based data [43]. Especially in case of a medium-sized cochlea with a CDL of 33–35 mm, small measurement-related differences in the CDL due to different radiological methods used might affect the selection of the electrode array.

Studies on comparative CDL measurements have not yet addressed the impact of the measures on electrode array selection. Using the cochlear length at 720° for electrode selection, as recommended by the manufacturer, resulted in an agreement in electrode array selection in 90.17% of all cases by comparing CT-based and MRI-based measures. As none of the cochlear lengths at two turns were less than 28 mm, only FLEX 28 and electrode arrays with a length of 31.5 mm were determined as the most suitable electrode array. Based solely on the CDL measurement, the long electrode array of 31.5 mm was considered optimal in 67.1% of the CT-based measurements and in 65.4% of the MRI-based measurements. In the remaining cases, a FLEX 28 electrode array was found to be the best choice.

According to the aforementioned studies, the AID plays a significant role in the postoperative gain in speech perception in recipients with lateral wall electrode arrays. The mean AID achieved with the electrode array which was considered to be optimal was 634.2° (range: 543.0°–701.9°) based on the CT scan and 630.7° (range: 535.6°–682.4°) based on the MRI scan. The slightly smaller AID in MRI-based measurements might be due to the slightly smaller portion of long 31.5 mm electrode arrays in this group. However, a serial comparison of the AID of the most appropriate electrode array did not show a significant difference between the two radiological procedures, even though a 31.5 mm array was chosen less frequently in MRI-based measurements.

The present study shows that despite an individually adapted selection of the electrode array from the currently available lateral wall electrode array portfolio of longer CI electrode arrays, complete cochlear coverage of 80% [45] was not achieved. Even though the present study only deals with virtually calculated electrode array insertion depths, similar results have been found in studies analysing the insertion depth of physically inserted electrode arrays. As mentioned by Spiegel et al., who compared the electrode array insertion length of the FLEX 28 and the 31.5 mm arrays in 378 implanted ears, an AID of two turns of the cochlea was not reached in any case [51]. A comparison of different methods for the measurement of the CDL based on CT-data in human temporal bones implanted with a 31.5 mm CI electrode array showed an underestimation of the CDL. This means that all measurements might lead to the selection of electrode arrays that are too short [36]. Thus, to obtain the highest possible percentage of cochlear coverage with a lateral wall electrode array, the longer 31.5 mm electrode array should be selected already from a length at two turns of about 30.5–31.5 mm.

In addition to the impact of different radiological diagnostic procedures, the fact that interindividual rater-related differences might also influence the selection of the optimal electrode array must also be taken into account when assessing the reliability of radiological measurement of the CDL. In previous studies, comparative evaluations of CDL measurements using Otoplan were mostly performed by two raters based on CT scans [32,36,37,55,56] and in one case based on an MRI scan [53].

In the present study, the interrater reliability of the CDL based on CT measurements was very high, with an intercorrelation coefficient (ICC) of 0.879–0.895. This is higher than reported by Cooperman et al., who described an interrater reliability with an ICC of 0.54 [56]. However, Breitsprecher et al. showed an even better interrater reliability, with an ICC of 0.94 between two raters [36]. This was in line with Canfarotta et al., which was the first study dealing with the reliability of Otoplan results based on CT data. They found an interrater reliability of the AID of the inserted electrode array with an ICC of 0.980 in a postoperative comparative study. Data allowing a direct comparison with our data on the interrater reliability of the CDL were not available in this study [55].

The mean interrater difference of the CDL in the present study was 0.64–0.86 mm and the interrater reliability was 0.789–0.851 based on the MRI data, which can be considered excellent. The only study so far published on this topic showed an even lower mean interrater difference in MRI data of 0.15 mm with a good-to-nearly-excellent interrater reliability of ICC 0.73 [53].

However, despite an overall good-to-very-good agreement in the measurement of different raters using both CT data and MRI data, the same electrode array was determined only in 61.5–66.7% of the CT-based measurements and in 59.0–61.5% of the MRI-based measurements. The basis for determining the individually optimal electrode array was the cochlear length at 720°.

A lack of consistency in terms of uniform electrode array selection was found particularly in medium-sized cochleae, with a CDL of approximately 33–35 mm. Considering the recently published data which clearly demonstrate a radiological underestimation of the cochlear length parameters, it seems reasonable to select a longer electrode array (31.5 mm) in these cases to obtain the highest percentage of cochlear coverage with a lateral wall electrode array.

In the future, further improvements in the accuracy of CDL measurement might be achieved by an automatic approach. First studies on automatic CDL measurements using CT-based data are promising and showed more replicable and less time-consuming results [57]. However, studies must clarify whether this is also possible with MRI-based data.

The present study is not without limitations. As already mentioned, we only dealt with a virtual appraisal of the achieved AID of the most appropriate electrode array calculated by Otoplan, since the study was based on preoperative radiological examinations. A postoperative MRI with the CI electrode array in place did not seem reasonable purely for scientific reasons. Therefore, the real AID might be lower than the virtually calculated one. The uniform determination of the AID of the inserted electrode arrays within the standard postoperative radiological control by means of a HR-CT was not possible in the present study because cochlear implantation was not exclusively performed with lateral wall electrode arrays of the manufacturer MED-EL^®^ but also with perimodiolar electrodes of other manufacturers. Since the measurements were performed with Otoplan version 2.0, updates to version 3.0 were not included in the study. Thus, further developments of the otosurgical planning system and their possible influence on the uniformity of the CDL evaluations must be factored in.

## 5. Conclusions

Preoperative CDL measurement using Otoplan, which facilitates personalized fitting of CI candidates with the most appropriate electrode array, showed no significant difference between MRI- based and CT-based data and had a very high intrarater reliability and low intrarater differences. Differences due to the radiological methods used were equal or even smaller than rater-related differences. An intensive training period might help to reduce the intra- and interrater differences, also in MRI-based measurements. However, differences of about 3 mm may result in significant inconsistencies in electrode array selection. Therefore, from a threshold value of about 30.5–31.5 mm at 720°, a 31.5 mm lateral wall electrode array might be the electrode of choice to obtain the highest percentage of cochlear coverage.

## Figures and Tables

**Figure 1 jpm-13-00633-f001:**
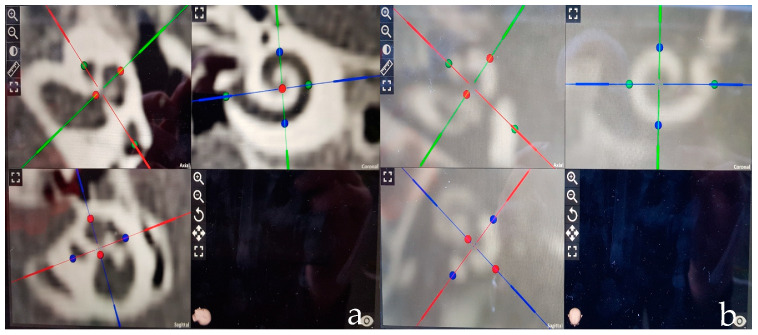
Measurement of cochlear parameters (diameter A = green, width B = blue, and height H = red) of a left cochlea as shown by the tablet-based Otoplan 2.0 software based on (**a**) CT data as well as (**b**) MRI data.

**Figure 2 jpm-13-00633-f002:**
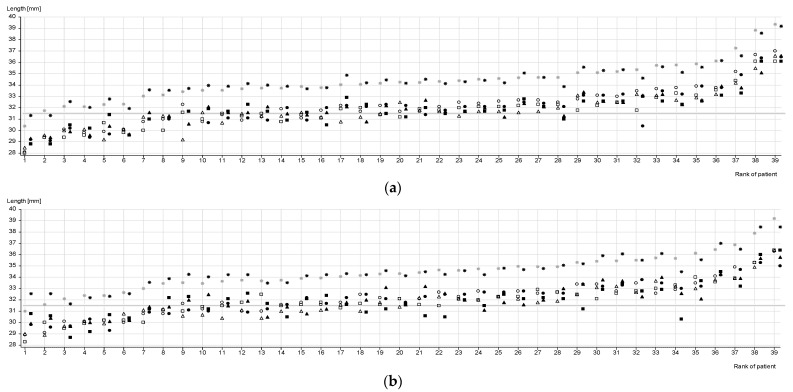
Measurements of the cochlear duct length (CDL) and length at 720° of each cochlea based on CT and MRI of (**a**) left cochleae (*n* = 39) and (**b**) right cochleae (*n* = 39). Blank white symbols indicate CT-based data, filled black symbols indicate MRI-based data; ✵, ✸= mean CDLs evaluated by all 3 raters, ○, ● = lengths at 720° evaluated by rater 1, □, ■ = lengths at 720° evaluated by rater 2, ∆, ▲ = lengths at 720° evaluated by rater 3.

**Figure 3 jpm-13-00633-f003:**
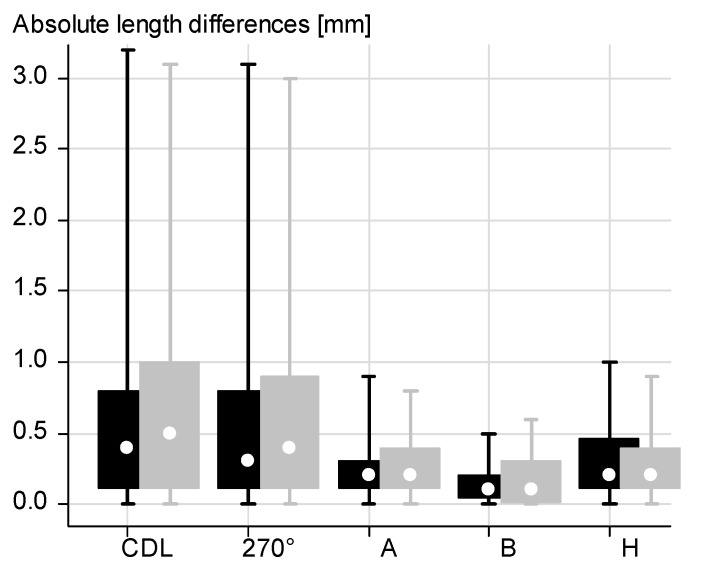
Absolute intrarater differences of length parameters based on CT vs. MRI for the left (black) and right ear (grey), CDL= cochlear duct length, 270° = length at 720°, A = length of diameter A, B = length of width B, H = length of height H.

**Figure 4 jpm-13-00633-f004:**
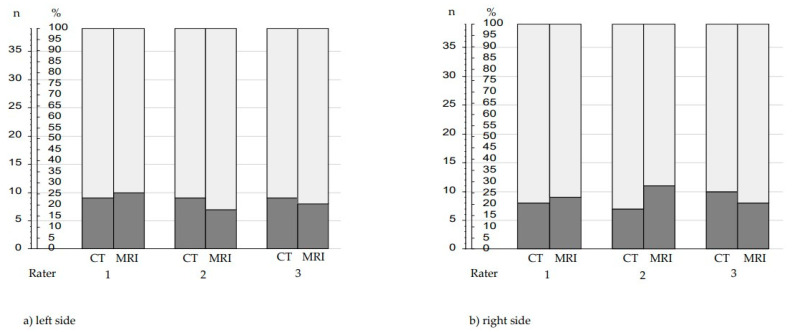
Electrode arrays (FLEX 28 vs. 31.5 mm electrode array) as suggested by the 3 raters based on CT vs. MRT data in (**a**) left or (**b**) right cochleae; light grey = number of 31.5 mm electrode arrays; dark grey = number of FLEX 28 electrode arrays.

**Table 1 jpm-13-00633-t001:** Length of the cochlea (CDL = cochlear duct length, length at 720° = cochlear length measured at 720° (2 turns), Diameter A = transverse diameter A of the basal turn of the cochlea, Width B = transverse width B of the basal turn of the cochlea, Height H = height of the cochlea), mean ± standard deviation and (range) of absolute lengths [mm].

	Rater 1	Rater 1	Rater 2	Rater 2	Rater 3	Rater 3
	CT	MRI	CT	MRI	CT	MRI
**Right cochlea**						
CDL	34.628 ± 1.687 (31.2–39.1)	34.641 ± 1.595 (31.5–38.1)	34.426 ± 1.655 (30.5–39.2)	34.392 ± 1.672 (31.0–39.2)	34.264 ± 1.667 (31.1–39.3)	34.59 ± 1.596 (32.0–38.5)
Length at 720°	32.136 ± 1.567 (28.9–36.3)	32.151 ± 1.482 (29.3–35.3)	31.946 ± 1.53 (28.3–36.4)	31.918 ± 1.548 (28.7–36.4)	31.792 ± 1.545 (28.9–36.4)	32.11 ± 1.481 (29.7–35.8)
Diameter A	8.913 ± 0.397 (8.0–9.7)	9.021 ± 0.418 (8.3–10.0)	8.864 ± 0.324 (8.3–9.6)	8.787 ± 0.379 (8.1–9.8)	8.913 ± 0.388 (7.9–9.8)	8.882 ± 0.389 (8.2–9.8)
Width B	6.81 ± 0.345 (6.0–7.7)	6.777 ± 0.301 (6.1–7,3)	6.774 ± 0.372 (5.9–7.9)	6.795 ± 0.348 (6.2–7.9)	6.718 ± 0.342 (6.0–7.7)	6.81 ± 0.309 (6.3–7.6)
Height H	3.738 ± 0.318 3.0–4.4	3.9 ± 0.298 (3.1–4.5)	3.733 ± 0.267 (3.1–4.3)	3.803 ± 0.33 (3.2–4.8)	3.597 ± 0.27 (3.2–4.1)	3.618 ± 0.263 (3.1–4.2)
**Left cochlea**						
CDL	34.644 ± 1.915 (30.3–39.8)	34.436 ± 1.831 (31.3–39.3)	34.195 ± 1.738 (30.2–38.9)	34.341 ± 1.563 (31.0–38.9)	34.177 ± 1.712 (30.7–39.4)	34.382 ± 1.592 (31.5–39.4)
Length at 720°	32.156 ± 1.794 (28.1–37.0)	31.972 ± 1.71 (29.1–36.5)	31.746 ± 1.617 (28.0–36.1)	31.874 ± 1.456 (28.8–36.1)	31.728 ± 1.59 (28.5–36.6)	31.923 ± 1.473 (29.3–36.6)
Diameter A	8.938 ± 0.335 (8.2–9.5)	8.9 ± 0.384 (8.2–9.8)	8.895 ± 0.329 (8.1–9.4)	8.856 ± 0.376 (8.3–9.7)	8.91 ± 0.34 (8.2–9.4)	8.892 ± 0.365 (8.1–9.6)
Width B	6.805 ± 0.452 (5.6–8.0)	6.787 ± 0.361 (6.1–7.8)	6.721 ± 0.411 (5.6–7.8)	6.754 ± 0.323 (6.0–7.7)	6.695 ± 0.405 (5.8–8.0)	6.751 ± 0.352 (6.0–7.8)
Height H	3.823 ± 0.346 (3.1–4.3)	4.021 ± 0.318 (3.4–4.6)	3.744 ± 0.312 (3.1–4.3)	3.856 ± 0.339 (2.8–4.5)	3.644 ± 0.279 (3.0–4.1)	3.662 ± 0.248 (3.0–4.1)

**Table 2 jpm-13-00633-t002:** Number of electrode arrays selected as the most appropriate by the 3 raters (Rater columns: 31 = 31.5 mm electrode array selected, 28= FLEX 28 electrode array selected).

			Left Side				Right Side			
Rater			HR-CT		MRI		HR-CT		MRI	
1	2	3	n	%	n	%	n	%	n	%
31	31	31	17	43.6	16	41.0	19	48.7	17	43.6
31	31	28	6	15.4	3	7.7	8	20.5	3	7.7
31	28	31	4	10.2	3	7.7	2	5.1	6	15.4
31	28	28	3	7.7	1	2.6	2	5.1	1	2.5
28	31	31	1	2.6	6	15.4	0	0	2	5.1
28	31	28	0	0	2	5.1	1	2.6	2	5.1
28	28	31	1	2.6	1	2.6	0	0	1	2.6
28	28	28	7	17.9	7	17.9	7	18.0	7	18.0

## Data Availability

The data presented in this study are available on request from the corresponding author.

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
