# Peer review of "Intra- and Interrater Reliability of CT- versus MRI-Based Cochlear Duct Length Measurement in Pediatric Cochlear Implant Candidates and Its Impact on Personalized Electrode Array Selection"

_jpm, 2023, doi:10.3390/jpm13040633_

Round 1
Reviewer 1 Report
Thank you very much for the opportunity to review the manuscript titled "Intra- and interrater reliability of CT- versus MRU-based cochlear duct length measurement in pediatric cochlear implant candidates and its impact on personalized electrode array selection" for JPM. In this study, authors aimed to evaluate the MRI and CT scans, in order to compare the measurements obtained in each method and if it impacts on electrode array choice. This is a very interesting study, despite the fact that many different studies have performed similar comparisons, however, using CT scans. The applicability of MRI scans is yet to be validated. I'd like to suggest to add some figures in order to explain the method of measurement.
Author Response
Dear Sir or Madam,
thank you very much for your review of our paper. Below you will find an itemized, point-by-point response to the comments of the reviewers:
Reviewer 1
Thank you very much for the opportunity to review the manuscript titled "Intra- and interrater reliability of CT- versus MRU-based cochlear duct length measurement in pediatric cochlear implant candidates and its impact on personalized electrode array selection" for JPM. In this study, authors aimed to evaluate the MRI and CT scans, in order to compare the measurements obtained in each method and if it impacts on electrode array choice. This is a very interesting study, despite the fact that many different studies have performed similar comparisons, however, using CT scans. The applicability of MRI scans is yet to be validated. I'd like to suggest to add some figures in order to explain the method of measurement
According to your suggestion we added Figure 1 which depicts the comparison of CT- versus MRI-based length measurement methods of cochlear length parameters by using the Otoplan 2.0 software.
Reviewer 2 Report
The author measures the length of the cochlea through the results of imaging examination, but considering the resolution and error rate of imaging examination, the accuracy of the results will be affected. It is suggested that the author explain the resolution accuracy of the image system.
In this way, readers can have a better judgment on the final result
Author Response
Dear Sir or Madam,
thank you very much for your review of our paper. Below you will find an itemized, point-by-point response to the comments of the reviewers:
Reviewer 2
The author measures the length of the cochlea through the results of imaging examination, but considering the resolution and error rate of imaging examination, the accuracy of the results will be affected. It is suggested that the author explain the resolution accuracy of the image system.In this way, readers can have a better judgment on the final result.
Thank you very much for this important comment and suggestion. Differences in the resolution accuracy of the two different radiological imaging techniques might certainly have an impact on the results of the length measurements of the cochlear parameters. The investigation and impact of this aspect is a main question of this study. However, after consultation with the manufacturer of the radiological systems used in the study according to his statement a direct comparison of the resolution accuracy of high-resolution CT vs. MRI which were used in the present study is not possible. CT has the advantage of high spatial resolution and multiplanar capabilities, whereas MRI has greater contrast resolution than CT. To emphasize this point, we have added the following topic to the discussion to line 393: ”However, a clear statement about the impact of differences of the resolution accuracy of the radiological procedures on the determination of the length parameters is not possible. CT has the advantage of high spatial resolution and multiplanar capabilities, whereas MRI has greater contrast resolution than CT.”